# Predicting Cognitive Decline in Motoric Cognitive Risk Syndrome Using Machine Learning Approaches

**DOI:** 10.3390/diagnostics15111338

**Published:** 2025-05-26

**Authors:** Jin-Siang Shaw, Ming-Xuan Xu, Fang-Yu Cheng, Pei-Hao Chen

**Affiliations:** 1Institute of Mechatronic Engineering, National Taipei University of Technology, Taipei 106, Taiwan; jshaw@ntut.edu.tw (J.-S.S.); ja99181@gmail.com (M.-X.X.); 2Institute of Long-Term Care, MacKay Medical College, New Taipei City 252, Taiwan; fycheng@mmc.edu.tw; 3Dementia Prevention and Treatment Center, MacKay Memorial Hospital, Taipei 104, Taiwan; 4Department of Medicine, MacKay Medical College, New Taipei City 252, Taiwan

**Keywords:** motoric cognitive risk syndrome, cognitive change, gait disorders, machine learning, support vector machine, feature ranking

## Abstract

**Background**: Motoric Cognitive Risk Syndrome (MCR), defined by the co-occurrence of subjective cognitive complaints and slow gait, is recognized as a preclinical risk state for cognitive decline. However, not all individuals with MCR experience cognitive deterioration, making early and individualized prediction critical. **Methods**: This study included 80 participants aged 60 and older with MCR who underwent baseline assessments including plasma biomarkers (β-amyloid, tau), dual-task gait measurements, and neuropsychological tests. Participants were followed for one year to monitor cognitive changes. Support Vector Machine (SVM) classifiers with different kernel functions were trained to predict cognitive decline. Feature importance was evaluated using the weight coefficients of a linear SVM. **Results**: Key predictors of cognitive decline included plasma β-amyloid and tau concentrations, gait features from dual-task conditions, and memory performance scores (e.g., California Verbal Learning Test). The best-performing model used a linear kernel with 30 selected features, achieving 88.2% accuracy and an AUC of 83.7% on the test set. Cross-validation yielded an average accuracy of 95.3% and an AUC of 99.6%. **Conclusions**: This study demonstrates the feasibility of combining biomarker, motor, and cognitive assessments in a machine learning framework to predict short-term cognitive decline in individuals with MCR. The findings support the potential clinical utility of such models but also underscore the need for external validation.

## 1. Introduction

Motoric Cognitive Risk Syndrome (MCR) is a preclinical condition characterized by the co-occurrence of subjective cognitive complaints and slow gait. It has been associated with an increased risk of developing dementia and is considered a valuable screening construct in aging populations [1,2,3]. Compared to Mild Cognitive Impairment (MCI), MCR is simpler to diagnose and does not require objective evidence of cognitive dysfunction, making it more feasible in primary care and community settings [4].

Both components of MCR—slow gait and subjective cognitive impairment—have independently been linked to adverse neurological outcomes. Gait slowing has also been linked to cognitive decline, with studies highlighting its association with reduced brain volume in motor and cognitive regions, including the prefrontal cortex [5,6]. Subjective cognitive impairment (SCI), characterized by an individual’s perception of memory or other cognitive decline without objective cognitive deficits, predicts subsequent dementia risk [7,8]. Their combination in MCR may therefore reflect an early stage of neurodegenerative processes. However, longitudinal studies show that not all individuals with MCR experience cognitive deterioration; some remain stable or even improve [9]. This heterogeneity presents a clinical challenge and underscores the need for tools that can identify which individuals with MCR are at highest risk of decline.

Most prior studies have explored predictors of decline in MCR using single-domain measures (e.g., gait, cognition, or biomarkers alone), and often rely on cross-sectional analyses [10,11,12]. Although Verghese et al. [3] introduced MCR as a screening construct, few studies have systematically tested how multimodal data could be integrated to improve prognostic accuracy.

Traditional methods of identifying at-risk individuals, such as clinical assessments and simple observational studies, often need more precision to capture the complexity of MCR and its progression. This gap calls for integrating more advanced analytical approaches, such as machine learning models, to provide a more comprehensive understanding of MCR and its predictive factors for cognitive decline. Machine learning models are crucial in this context, as they can analyze complex, multidimensional data to uncover patterns that traditional methods might miss.

Machine learning techniques have emerged as powerful tools in medical research, offering superior predictive accuracy and the ability to manage large-scale, complex datasets. These methods have been widely applied in prognostic studies of diseases such as dementia [11]. Among them, Support Vector Machines (SVMs) have demonstrated particular suitability in the medical domain due to their effectiveness in handling high-dimensional and small-sample data [13,14]. Similarly, Random Forests have shown promising performance and share the advantage of providing feature importance rankings [15,16]. Such capabilities are instrumental in identifying key variables associated with cognitive decline. By leveraging these machine learning approaches, it becomes possible to enhance the early identification of high-risk individuals, thereby facilitating more precise prevention and intervention strategies.

To address this gap, we developed a machine learning model to predict one-year cognitive decline in older adults with MCR by integrating neurodegenerative biomarkers (e.g., β-amyloid, tau), dual-task gait metrics, and neuropsychological assessments. Our central research question is as follows: Can multimodal data improve the short-term prediction of cognitive decline in MCR using SVM models compared to single-domain predictors? We hypothesize that combining biological, motor, and cognitive information will significantly improve prediction performance, offering a clinically useful tool for early risk stratification.

## 2. Materials and Methods

### 2.1. Proposed Approach

The system framework for building a machine learning model to classify whether MCR patients will experience cognitive decline after one year is shown in Figure 1. All participants must be diagnosed with MCR. It is important to note that some participants may be unable to perform specific physical tasks, such as jumping, due to knee injuries. In contrast, others may have missing data due to equipment malfunctions. These missing data were imputed using Bayesian ridge regression. After one year, participants were diagnosed by neurologists and classified as either cognitively declined or non-cognitively declined. The initial clinical data, various biomarkers, gait parameters, and cognitive function were then used to train the SVM classification model. Finally, the model’s predictive accuracy (ACC) was evaluated, feature importance was ranked based on the model’s weight coefficients, and 5-fold cross-validation was conducted to validate the model’s performance.

### 2.2. Participants

The inclusion criteria for study participants were as follows:Age 60 years or older;Ability to walk independently for more than 10 m without assistive devices;Presence of subjective cognitive complaints (as confirmed by answering “yes” to the memory-related question on the Geriatric Depression Scale [17]: “Do you feel you have more memory problems than most people?”);Slow gait speed, defined as being more than 1.5 standard deviations below the age-adjusted norm.

The exclusion criteria were as follows:
Diagnosed dementia;Severe depression;Neurological diseases;Any other conditions that might interfere with participation in the study.

### 2.3. Data Analysis

This study utilizes a dataset comprising 80 samples, each with 61 features. The data originates from physiological measurements of MCR participants and is significantly related to MCR factors [10,18]. This dataset includes the following:Clinical data (9 features): Heart Disease, Visual Impairment, Number of Falls, Left Hand Grip Strength (kg), BMI (Body Mass Index, kg/m^2^), Neurological Disease, Age (years), Years of Education, Total Number of Diseases, Gender (male/female), Hypertension, Right Hand Grip Strength (kg), Diabetes.Plasma neurodegenerative biomarkers (2 features): Abeta1-42 Mean (pg/mL), Tau Mean (pg/mL).Gait parameters (30 features): TUG_StandToSit (cognitive task, seconds), TUG_MidTurn (motor task, seconds), Cadence (cognitive task, steps/min), TUG_SitToStand (motor task, seconds), Cadence (steps/min), Left Stride Duration (cognitive task, seconds), TUG_Return (cognitive task, seconds), TUG_Return (motor task, seconds), TUG_EndTurn (motor task, seconds), TUG_Forward (cognitive task, seconds), Speed (meters/second), TUG_MidTurn (cognitive task, seconds), Speed (cognitive task, meters/second), Five-Meter Walk (seconds), Stride Length (meters), TUG Time (cognitive task, seconds), TUG_Forward (motor task, seconds), TUG Time (motor task, seconds), Five-Time Sit-to-Stand, Balance Test, Cadence (motor task, steps/min), Stride Length (cognitive task, meters), Stride Length (motor task, meters), TUG_StandToSit (motor task, seconds), Speed (motor task, meters/second), TUG_SitToStand (cognitive task, seconds).Cognitive function metrics (20 features): Judgement of Line Orientation, CVLT-SF (California Verbal Learning Test—Short Form), GDS-15 (Geriatric Depression Scale, 15 items), TMT-B Correct Number, TMT-A (seconds), CVLT-SF-30S (30 s delay recall), CVLT-SF-10MIN (10 min delay recall), CVLT-SF-cued, Digit Symbol, TMT-B (seconds), MOCA (Montreal Cognitive Assessment), Boston Naming (Correct + Cued Correct), MMSE (Mini-Mental State Examination), Boston Naming, TMT-A Correct Number, FRIED Frailty Classification.

This dataset captures cognitive decline in older adults with MCR over a one-year follow-up period. Among the 80 participants, 16 experienced cognitive decline, representing 20% of the cohort. Consequently, the dataset exhibits class imbalance.

Missing data were imputed using the Bayesian Ridge regression method. During data preprocessing, standardization was applied, and the dataset was split into training and testing sets in a 4:1 ratio. To address class imbalance and prevent model bias due to the underrepresentation of the minority class, the Synthetic Minority Over-sampling Technique (SMOTE) was employed to increase the minority class number. Detailed description was given in Section 2.3.2.

A SVM classification model was developed using scikit-learn version 1.5.0, with hyperparameters optimized through GridSearchCV. Feature importance was also analyzed. Given the limited number of minority-class samples and the dataset imbalance, k-fold cross-validation was implemented to ensure model stability. The model’s performance was primarily evaluated using the Area Under the Curve (AUC) metric.

#### 2.3.1. Imputation of Missing Data

According to a comparative example provided in the official scikit-learn documentation [19], Bayesian ridge regression showed relatively stable and robust imputation performance across different scenarios, outperforming other alternatives such as Random Forest and KNN in the evaluated synthetic dataset. Therefore, we selected Bayesian Ridge as the imputation method in this study.

Bayesian ridge regression is a linear regression model that combines Ridge Regression and Bayesian inference [20,21,22]. This approach introduces L2 regularization by placing a Gaussian prior on the regression coefficients. The model estimates the posterior distribution of the coefficients β, which can also be used to impute missing values based on the inferred distribution [22].

The posterior distribution of the coefficients, given the observed data X and target values y, is expressed using Bayes’ theorem as:(1)$$p\left(\beta |X,y\right)=\frac{p\left(y|X,\beta \right)*p\left(\beta \right)}{p\left(y|X\right)}$$

The key components of this formulation include:Likelihood function $p\left(y|X,\beta \right)$: This denotes the probability of observing y given X and β, assuming a Gaussian distribution with mean Xβ and covariance $\alpha^{-1}$. Here, $\alpha^{-1}$ represents the observation noise variance. A smaller variance (i.e., larger α) implies that the data are tightly concentrated around the model’s predictions:



(2)
$$p\left(y|X,\beta \right)∼N\left(X\beta ,\alpha^{-1}I\right)$$



Prior distribution function $p\left(\beta \right)$: This expresses prior belief about β, modeled as a zero-mean Gaussian with covariance $\lambda^{-1}$. A larger precision λ (i.e., smaller prior variance) imposes stronger regularization, effectively shrinking the coefficients toward zero:



(3)
$$p\left(\beta \right)∼N\left(0,\lambda^{-1}I\right)$$



Marginal likelihood function $p\left(y|X\right)$: Also known as the model evidence, this term normalizes the posterior distribution and ensures it integrates into one.

Bayesian ridge regression not only provides point estimates of the model parameters but also quantifies their uncertainty. Additionally, the hyperparameters α and λ are typically estimated from the data, allowing the model to adaptively control the regularization strength. This mechanism helps to prevent overfitting and enhances model robustness, especially in the presence of multicollinearity or small sample sizes [23].

#### 2.3.2. Data Preprocessing

First, feature scaling is required to ensure that each feature has the same range, preventing large numerical differences from negatively affecting model performance [24,25]. Using min–max normalization, the original feature x is scaled to a value $x^{\prime }$ within the range [0, 1], without altering the intrinsic characteristics of the original dataset:(4)$$x^{\prime }=\frac{x-x_{\min}}{x_{\max}-x_{\min}}$$

Here, $x_{\max}$ and $x_{\min}$ represent the peak and bottom values of the feature.

Due to the imbalance in the dataset, with a ratio of approximately 1:4 between cognitive decline and non-decline cases, this imbalance could lead to biased model performance. To address this, SMOTE is employed to increase the number of samples in the minority class of cognitive decline [26,27,28].

Assuming that x belongs to the minority class set A, SMOTE generates new samples through the following steps:

Using Euclidean distance, for each minority class sample $x_{i}$, find its K nearest neighbors $x_{i}^{\mathrm{nn}}$ within A.

Randomly select a sample $x_{i}^{\mathrm{nn}}$ from these K neighbors and generate a new sample using linear interpolation:(5)$$x_{\mathrm{new}}=x_{i}+\delta \cdot \left(x_{i}^{\mathrm{nn}}-x_{i}\right)$$
where *δ* ∈ [0, 1] is a randomly chosen weight coefficient to control the position of the new sample.

The newly generated sample $x_{\mathrm{new}}$ is added to the minority class dataset. This process is repeated until the minority class reaches the desired sample size. The data distribution after SMOTE preprocessing was presented in Table 1.

#### 2.3.3. GridSearchCV Approach

GridSearchCV is a hyperparameter optimization technique that systematically explores a predefined set of hyperparameters through an exhaustive search combined with cross-validation to identify the optimal parameter configuration [29,30]. In this study, GridSearchCV was utilized to optimize the different kernels of the SVM. Specifically, for the linear kernel, the C parameter was tuned; for the radial basis function (RBF) kernel, both C and gamma were optimized; and for the polynomial (Poly) kernel, C, gamma, and degree parameters were included in the search. Given the imbalanced nature of the dataset, the area under the receiver operating characteristic curve (AUC) was employed as the primary evaluation metric to determine the optimal hyperparameter settings for each kernel [31].

#### 2.3.4. SVM

SVM is a supervised learning method designed to map data into a high-dimensional feature space, aiming to find a hyperplane that best separates two classes. The margin between the two classes on this hyperplane is referred to as the decision boundary, and the objective of SVM is to maximize this boundary. The shape of the decision boundary is determined by various kernel functions, which allow for the construction of linear or non-linear classification models [32]. SVM has been proven to perform well in various medical diagnostic tasks [13,33,34,35].

In this study, different kernel functions—including linear, radial basis function (RBF), and polynomial (Poly) kernels—were employed to compare which model performs best in predicting cognitive decline. The linear kernel assumes a linearly separable structure and is suitable when the number of features is large relative to the number of samples. The RBF kernel is widely used for capturing complex, non-linear relationships in the data. The polynomial kernel, which introduces interaction terms of varying degrees, is often used when the data exhibit moderate non-linear patterns while preserving interpretability [36].

#### 2.3.5. Feature Ranking

The performance of SVM largely depends on the correct selection of features. Effective feature selection not only reduces the computational complexity of the model but also mitigates the risk of overfitting [37]. In this study, the linear kernel of the SVM model was used to compute the weight distribution for each feature, denoted as w, representing the influence of each feature on the orientation of the hyperplane. By taking the absolute values of w and ranking them in descending order, the importance of features in the SVM linear model can be determined. This ranking allows us to assess feature significance [38,39], which is crucial for identifying key features associated with the decline in MCR functionality.

It is worth noting that this weight-based method of feature importance is only applicable to linear kernels, as non-linear kernels such as RBF or polynomial do not produce explicit weight vectors.

#### 2.3.6. K-Fold Cross-Validation

K-fold cross-validation is a method of partitioning a dataset into K equally sized subsets to enhance a model’s generalization ability. The data are split while preserving the proportion of each class. In each iteration of training, K − 1 subsets are used as the training set, while the remaining subset is used for validation. This process is repeated K times, ensuring that each subset serves as the validation set exactly once. The final model performance is obtained by averaging the results across all K iterations, reducing the potential bias caused by a single data split [40].

For example, in 5-fold cross-validation, if the dataset contains 100 samples, each iteration will allocate 80 samples for training and 20 samples for validation, repeating this process five times so that every sample is used for validation exactly once.

Cross-validation is particularly useful when dealing with limited datasets, as it mitigates misleading results caused by a single data split. It helps reduce variance, improves the reliability of evaluation metrics, and ensures that the model can generalize to different data distributions.

## 3. Results

In this study, an SVM model with a linear kernel was trained using 61 features. The model ranked feature importance based on weights, serving as a reference for subsequent feature selection. The goal was to identify the key subset of features most indicative of cognitive decline. Table 2 presented the top 30 features ranked by importance.

All subsequent experiments were conducted based on the feature importance rankings provided in Table 2. Starting with the top 5 features, one feature was added incrementally until the full set of 61 features was included. SVM models were trained using these different feature subsets and tested with three kernel functions: linear, RBF, and poly. The results are presented in Figure 2.

Due to the limited sample size in the dataset, we implemented 5-fold cross-validation to ensure model reliability. Two evaluation metrics were used: ACC in Figure 3 and AUC in Figure 4. Given the presence of class imbalance in the dataset, accuracy may not accurately reflect model performance. Therefore, AUC was primarily used as the key evaluation metric, while accuracy served as a secondary observation metric.

Based on the results shown in Figure 2, Figure 3 and Figure 4, the linear kernel demonstrated overall more stable performance compared to the poly and RBF kernels, with the poly kernel being moderately stable and the RBF kernel the least stable. Given its higher accuracy and stability, the linear kernel was adopted as the primary model structure. Moreover, feature importance ranking was derived from the linear kernel, reinforcing the consistency between model selection and feature evaluation.

In addition, Figure 4a shows that the linear kernel achieved relatively strong performance when selecting 28–32 features. Figure 4b indicates that the RBF kernel also performed best in the 30–32 feature range, while Figure 4c shows that the polynomial kernel achieved better performance with 30–35 features. Therefore, we focused on the shared range of 28–35 features as the potentially optimal interval for both adaptability and stability. The corresponding results of average performance metrics—accuracy, AUC, precision, recall, and F1 score—under 5-fold cross-validation within this feature range are presented in Table 3.

To further validate the model selection, a comparative analysis was conducted between the SVM and Random Forest (RF) classifiers. For the RF model, feature importance was derived from the mean decrease in impurity after training, with hyperparameters optimized using GridSearchCV. Both models utilized their own internal feature importance rankings to incrementally train models from the top 5 features up to all 61 features. Each configuration was evaluated using 5-fold cross-validation. As illustrated in Figure 5, the SVM consistently outperformed the RF classifier in both (a) ACC and (b) AUC, thereby supporting the choice of SVM as the primary classifier in this study.

Based on Table 3, although model performance under cross-validation was comparable across the 28–35 feature range, the model using 30 selected features achieved the highest AUC on the imbalanced test set—a metric prioritized in imbalanced classification tasks. We speculate that certain difficult-to-classify cases in the test set were more effectively distinguished when using exactly 30 features. Therefore, the final model was built using 30 features with a linear kernel. This model achieved a test accuracy of 88.2% and a test AUC of 83.7%, along with an average cross-validation accuracy of 95.3%, AUC of 99.6%, precision of 95.1%, recall of 94.9%, and F1-score of 95.7%.

## 4. Discussion

In this study, we utilized SMOTE to address class imbalance by synthetically augmenting the minority class, thereby aiming to enhance model performance and mitigate potential issues like overfitting often associated with imbalanced datasets. This method is particularly valuable in clinical research settings, where collecting sufficient minority class data—such as individuals experiencing cognitive decline—remains challenging. Feature selection based on linear SVM importance rankings enabled the identification of a compact and informative subset of predictors. This approach effectively mitigates the curse of dimensionality, especially in small-sample datasets, and enhances model performance.

Our choice of SVM as the primary classifier was further substantiated by a direct comparison with a Random Forest (RF) model. As detailed in the Results section and illustrated in Figure 5, the SVM demonstrated consistently superior performance over RF in both ACC and AUC across the incrementally evaluated feature subsets. This outcome is particularly significant for our study, given the context of a small and imbalanced dataset. In such scenarios, the SVM’s potential for robust performance and stability, along with the interpretability afforded by methods like linear SVM feature importance (as used in our feature selection process), can be critical advantages over ensemble techniques like RF.

Among the top 30 predictive features identified (Table 2), three categories emerged as highly influential: plasma biomarkers, gait parameters, and neuropsychological assessments. Plasma biomarkers, specifically elevated levels of β-amyloid and tau proteins, were consistently associated with cognitive decline, in line with known Alzheimer’s disease pathology. These results are broadly consistent with the findings of Zhuang et al. (2025) [10], who developed a time-dependent predictive model for Alzheimer’s disease using novel plasma protein signatures. While Zhuang et al. focused on longer-term dementia risk in a general aging population using seven protein biomarkers (e.g., APOE, NRCAM, CRP), our study complements this work by targeting a shorter prediction window (one year) and integrating functional data from gait and neuropsychological assessments, which may be more applicable in early-stage, clinically ambiguous cases such as MCR [10].

Gait parameters derived from the dual-task Timed Up and Go (TUG) test, including cadence and turnaround time, also demonstrated significant predictive value. These metrics reflect motor-cognitive integration and executive function—domains that may be subtly impaired during prodromal cognitive decline. The sensitivity of dual-task assessments to early functional changes supports their incorporation into clinical screening protocols.

Neuropsychological performance, particularly on the California Verbal Learning Test (CVLT), contributed meaningfully to prediction accuracy. As a standard measure of episodic memory, the CVLT captures early cognitive changes relevant to neurodegenerative processes. Incorporating such established tools enhances the model’s interpretability and practical utility.

Despite encouraging results, several limitations must be noted. The sample size was modest (*n* = 80), which may limit the generalizability of our findings. Although we employed 5-fold cross-validation to ensure model stability, future studies with larger, more diverse populations are needed to validate our results. Additionally, we did not include genetic markers such as APOE ε4 or neuroimaging data, which may provide added predictive power. Future studies should incorporate these modalities where feasible. Finally, while SMOTE effectively addresses class imbalance, it may introduce synthetic sample bias. Alternative balancing techniques, such as ADASYN or near-miss undersampling, could be explored in future research.

Future work should expand sample size and follow-up duration to model longitudinal cognitive trajectories more effectively. Integrating predictive tools such as ours into clinical workflows could facilitate early intervention and individualized care planning. However, clinical implementation requires prospective validation and assessment of model interpretability, especially in sensitive aging populations.

## 5. Conclusions

This study explored the use of interpretable machine learning methods to predict short-term cognitive decline in individuals with MCR. Given the inherent challenges of data imbalance, high-dimensional feature sets, and small sample sizes, we employed SMOTE to address class imbalance and used feature ranking to reduce dimensionality and identify potentially relevant predictors. A 5-fold cross-validation procedure was implemented to support internal model stability.

Our findings demonstrate that β-amyloid and Tau protein concentrations may serve as informative biomarkers associated with cognitive decline risk, consistent with established pathological mechanisms in early neurodegeneration. Gait characteristics, particularly those derived from the dual-task TUG test, also emerged as relevant features, likely reflecting early disruptions in motor-cognitive integration. While these findings are encouraging, they remain preliminary and must be interpreted with caution due to the modest sample size and lack of external validation. Future studies should aim to validate these findings in larger, more diverse cohorts to improve predictive performance. Additionally, longitudinal studies with extended follow-up are needed to examine the stability and clinical utility of machine learning–based risk models in real-world settings.

## Figures and Tables

**Figure 1 diagnostics-15-01338-f001:**
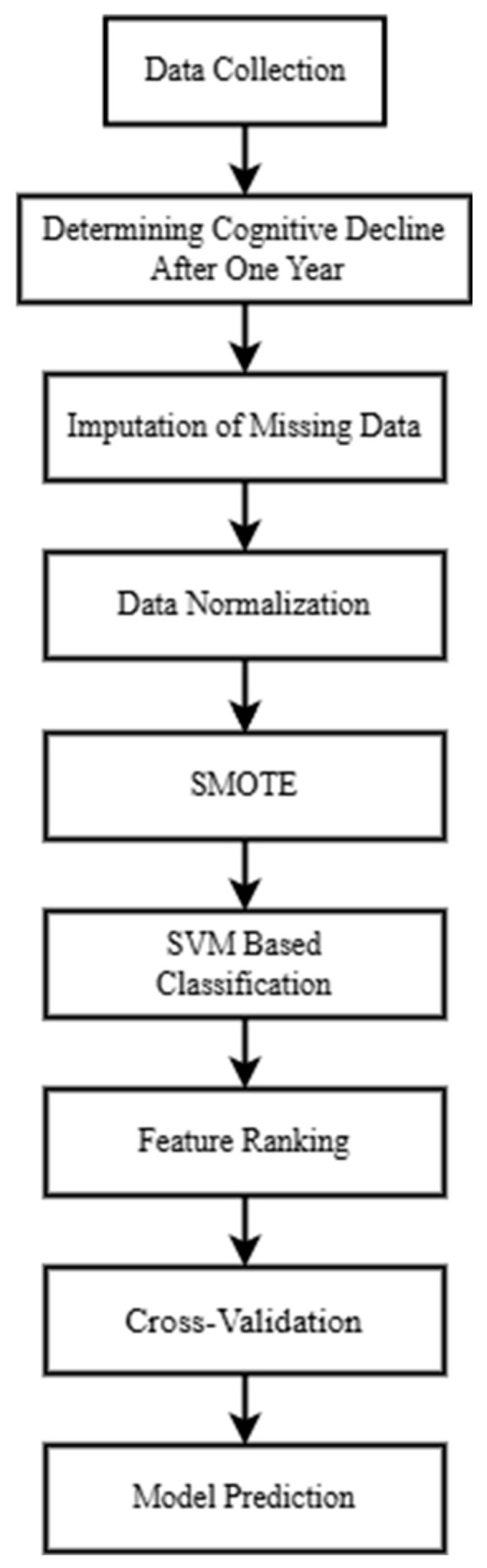
Proposed methodology framework.

**Figure 2 diagnostics-15-01338-f002:**
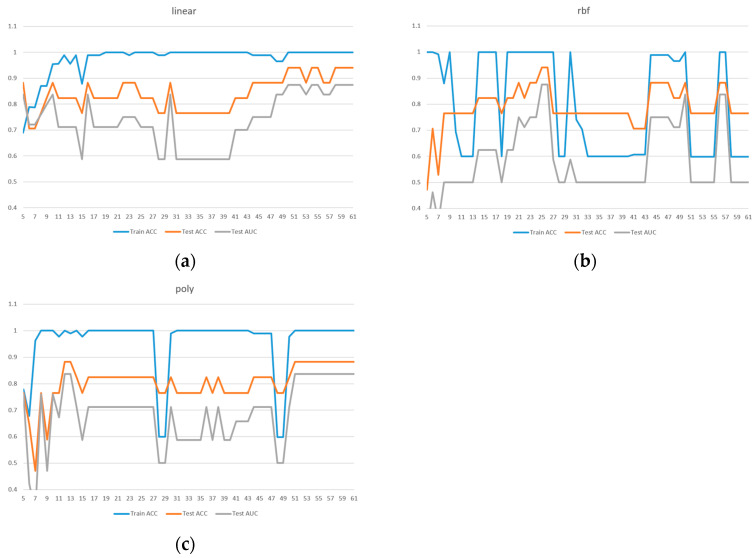
Comparison of model performance with different feature numbers for ACC, AUC: (**a**) linear kernel, (**b**) rbf kernel, (**c**) poly kernel.

**Figure 3 diagnostics-15-01338-f003:**
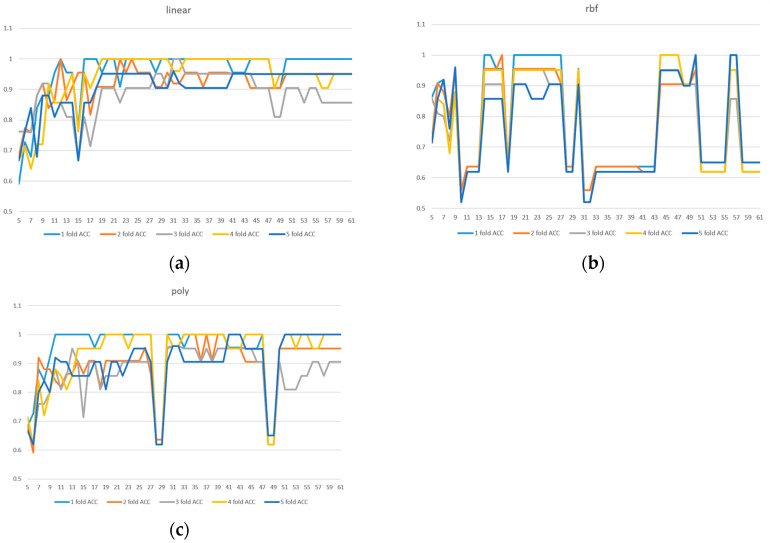
A 5-fold cross-validation comparison of model performance with different feature numbers using accuracy: (**a**) linear kernel, (**b**) rbf kernel, (**c**) poly kernel.

**Figure 4 diagnostics-15-01338-f004:**
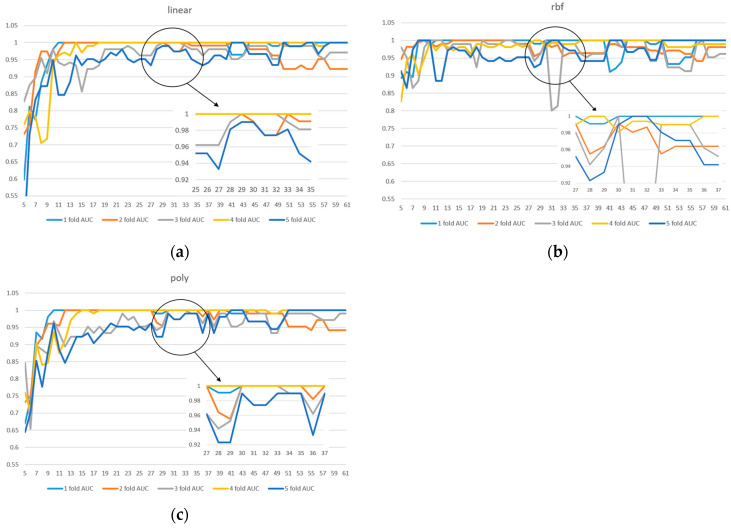
A 5-fold cross-validation comparison of model performance with different feature numbers using AUC: (**a**) linear kernel, (**b**) rbf kernel, (**c**) poly kernel.

**Figure 5 diagnostics-15-01338-f005:**
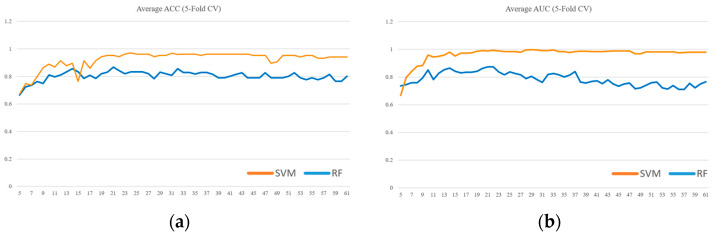
SVM vs. Random Forest performance across feature subsets evaluated by 5-fold cross-validation: (**a**) ACC, (**b**) AUC.

**Table 1 diagnostics-15-01338-t001:** Dataset before and after SMOTE.

		Train Data	Test Data
Before SMOTE	Non-cognitive decline	53	11
	Cognitive decline	11	5
After SMOTE	Non-cognitive decline	53	11
	Cognitive decline	35	5

**Table 2 diagnostics-15-01338-t002:** Feature importance ranking top 30.

Rank	Feature	Importance (w)
1	Judgement of Line Orientation	1.428
2	TUG_StandToSit (cognitive task, seconds)	−1.036
3	GDS-15 (Geriatric Depression Scale, 15 items)	1.025
4	Cadence (steps/min)	−0.884
5	Abeta1-42 Mean (pg/mL)	0.876
6	Visual Impairment	−0.853
7	TUG_SitToStand (motor task, seconds)	0.844
8	Cadence (cognitive task, steps/min)	−0.82
9	Tau Mean (pg/mL)	0.815
10	CVLT-SF-cued	0.729
11	Left Hand Grip Strength (kg)	0.72
12	Age (years)	0.71
13	TMT-B Correct Number	−0.704
14	CVLT-SF-30S (30 s delay recall)	−0.679
15	CVLT-SF (California Verbal Learning Test—Short Form)	−0.672
16	BMI (Body Mass Index, kg/m^2^)	−0.661
17	TUG_StandToSit (motor task, seconds)	0.645
18	Years of Education	0.642
19	Heart Disease	−0.614
20	Left Stride Duration (cognitive task, seconds)	0.612
21	TMT-A (seconds)	−0.595
22	TUG_Return (cognitive task, seconds)	0.591
23	TUG_MidTurn (motor task, seconds)	−0.542
24	Number of Falls	−0.519
25	TUG_Forward (cognitive task, seconds)	−0.511
26	Right Stride Duration (cognitive task, seconds)	0.502
27	CVLT-SF-10MIN (10 min delay recall)	−0.485
28	Gender (male/female)	−0.484
29	MMSE (Mini-Mental State Examination)	−0.457
30	Boston Naming (Correct + Cued Correct)	−0.432

**Table 3 diagnostics-15-01338-t003:** Detailed results of linear kernel feature selection under 5-fold cross-validation.

Feature Numbers	Test		5-Fold Cross-Validation				
	ACC	AUC	ACC	AUC	Precision	Recall	f1
28	0.765	0.587	0.944	0.994	0.941	0.942	0.945
29	0.765	0.587	0.953	0.998	0.951	0.949	0.958
30	0.882	0.837	0.953	0.996	0.951	0.949	0.957
31	0.765	0.587	0.968	0.989	0.968	0.97	0.97
32	0.765	0.587	0.96	0.989	0.96	0.961	0.961
33	0.765	0.587	0.962	0.994	0.96	0.962	0.96
34	0.765	0.587	0.962	0.984	0.96	0.962	0.96
35	0.765	0.587	0.962	0.982	0.96	0.962	0.96

## Data Availability

The datasets used in this study are not publicly available because of ethical restrictions preventing the public sharing of data.

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
