# Peer review of "Predicting Cognitive Decline in Motoric Cognitive Risk Syndrome Using Machine Learning Approaches"

_diagnostics, 2025, doi:10.3390/diagnostics15111338_

Round 1
Reviewer 1 Report
Comments and Suggestions for Authors
The paper called "Predicting Cognitive Decline in Motoric Cognitive Risk Syndrome Using Machine Learning Approaches" introduces a way to use machine learning, specifically Support Vector Machines (SVM), to evaluate the risk of cognitive decline in older adults with Motoric Cognitive Risk (MCR) syndrome. While the authors aim to address a timely and clinically significant problem, the manuscript suffers from several technical, methodological, interpretative, and presentation issues that collectively reduce the strength, novelty, and clinical applicability of the contribution.
Dense, poorly structured abstract. The abstract lacks clarity and utilizes technical language to explain the study's objectives and results. There is no explanation for a “remarkable increase” in performance metrics, and “integration of neurodegenerative biomarkers and motor-cognitive metrics effectively predicts cognitive decline” is a conclusion, not an abstract statement. The uneven use of words like “accuracy,” “AUC,” and “five-fold cross-validation” may mislead readers regarding the results' scope and trustworthiness.
Despite being anchored in relevant literature, the background and introduction are broad and do not focus on the research gap. The text repeats MCR and gait-cognition information without asking a new question. Prior work, such as Verghese et al. [9] or Bennett et al. [17], is cited but not rigorously reviewed to justify this study design. The introduction says that existing methods cannot represent MCR complexity precisely but does not explain how their machine learning solution overcomes these constraints.
The problem statement is ambiguous. No hypothesis and little justification exist for adopting SVM over other ML models. The text acknowledges Support Vector Machines in dementia studies but fails to distinguish their contribution from earlier models. Haller et al. (2010) and Javeed et al. (2022) employed machine learning on longitudinal cognitive datasets, but no comparison or critical review is provided.
The methodological description, while lengthy, lacks depth and clarity in several areas. The data imputation using Bayesian Ridge Regression is mathematically described, but the justification for choosing this method over others like multiple imputation or expectation-maximization is absent. Likewise, the method of using SMOTE to create more samples for balancing classes is used without checking for the potential problem of bias in the new samples, which is a common issue in high-dimensional data. The manuscript does not present alternatives such as ADASYN or ensemble balancing, nor does it attempt a sensitivity analysis of the impact of SMOTE on outcomes.
The model development pipeline is linear and lacks innovation. Feature scaling, hyperparameter tuning, and model evaluation follow standard textbook procedures without offering any optimization or insight. While GridSearchCV is a valid tuning method, it is computationally expensive and inefficient compared to more robust techniques like randomized search or Bayesian optimization, which could have been better suited given the small sample size. Moreover, using three SVM kernels without providing justification for their selection or describing their theoretical differences adds procedural bulk but no conceptual value.
The results section excessively relies on AUC and accuracy to report model performance, ignoring other critical metrics such as precision, recall, F1-score, or Matthews correlation coefficient. The reported 5-fold cross-validation AUC of 99.4% seems too high considering the small sample size (n=80) and suggests that the model may be too closely fitted to the training data, especially since it includes samples created by SMOTE. No external validation set or bootstrap resampling was used, which is a major limitation in any predictive modeling study, particularly one proposing clinical application.
Feature importance analysis is another weak point. The SVM weight-based ranking is appropriate only for linear kernels and does not generalize to non-linear models. The paper interprets weights as feature importance without correcting for multicollinearity, overfitting, or interaction terms. The highest-ranked features—such as Judgment of Line Orientation and TUG-related gait metrics—make sense in a clinical context, but the paper does not include effect sizes or statistical confidence intervals, which makes it difficult to evaluate how reliable these rankings are.
The discussion reiterates the results rather than expanding on their implications. It claims that the study integrates “dual-task assessments into routine clinical evaluations” without providing any evidence that these assessments were clinically implemented or validated. References to prior work are descriptive but lack comparative analysis. For example, the studies that used CVLT or tau protein to predict early cognitive decline (like Lagun et al., 2011; Zhuang et al., 2025) are not compared based on how well their methods work, how widely they can be applied, or their weaknesses. The discussion also fails to address the limitations of their modeling choice—SVM models are notoriously poor in interpretability and scalability for clinical deployment unless paired with understandable AI tools, which are not mentioned here. Refer to this article for further discussion: https://doi.org/10.3390/bioengineering10020249.
The conclusion was exaggerated without evidence. Explanations like “this study shows that β-amyloid and tau protein are significant biomarkers” are common. The tiny sample size and absence of external validation disprove the assertion that this study “confirms” their role. Without a live application or pilot, “machine learning can be used for early risk stratification” is speculative. The grammar is good; however, the manuscript is too technical and repetitious. The terms “robust indicators of cognitive decline” and “maximize diagnostic precision” are ambiguous.
Editing should focus on reducing redundancy, simplifying complex technical phrases, and enhancing clarity in the flow of methods and results.
Figures and tables are poorly presented. Figures 2–4 lack confidence intervals and error bars, making it impossible to judge performance variability. Table 2’s feature importance values are listed without statistical significance, and some features are ambiguously named (e.g., “Cadence (cognitive task)” vs. “Cadence”). Equations are accurate but unnecessary in their current length and depth. For example, the Bayesian Ridge formulas could be summarized rather than formally derived, as they do not contribute materially to understanding the model application.
To improve this study, the authors should:
Explicitly state the research question and hypothesis.
Compare their model to alternative ML algorithms (e.g., random forests, XGBoost).
Conduct external validation on an independent dataset.
Report a broader set of evaluation metrics with confidence intervals.
Reevaluate feature selection using permutation importance or SHAP values.
Provide practical implementation insights (e.g., app/tool prototype, integration into clinic).
Expand the literature review to include modern, understandable AI and longitudinal modeling approaches.
Refine the manuscript’s organization and streamline technical language.
Comments on the Quality of English LanguageThe language of the manuscript is grammatically sound but overly technical and redundant in many sections. Phrases such as “robust indicators of cognitive decline” and “maximize diagnostic precision” are vague and lack operational definition. Editing should focus on reducing redundancy, simplifying complex technical phrases, and enhancing clarity in the flow of methods and results.
Reviewer 2 Report
Comments and Suggestions for Authors
The paper presents a machine learning approach based on Support Vector Machines to predict cognitive decline in MCR patients. From a set of 61 initial features it applies feature importance metrics in order to manage accurate classification with as many as 30 features. From these, a discussion is opened about the feature categories that are more promising to serve as biomarkers for MCR. Data set imbalances are faced with SMOTE.
Very interesting and well presented research.
Some notes for consideration:
Explain the derivation of eq.(1) as P(b/X,y) would normaly be equal to [P(X,y / b)*P(b)]/P(X,y). The version you use make some assumptions that need to be explained. Maybe this part of the paper needs a slightly more detailed explanation, also regarding the terms a^{-1} etc.
Your approach of using SVM and then use different number of features according to their importance ranking is very interesting and it is the source of exploring the features contribution to the final decision. A related and similar approach on using machine learning (neural networks) and random decision forests for feature importance measures towards early diagnosis in the field of PTSD is also presented in https://doi.org/10.3390/app12157492
Round 2
Reviewer 1 Report
Comments and Suggestions for Authors
Authors are requested to revise their work in more depth as indicated by my previous comments. Their revision is not acceptable at its current form.
Round 3
Reviewer 1 Report
Comments and Suggestions for Authors
No further technical comments.
It was recommended to include more references.
Author Response
We have followed Reviewer 1’s suggestion and added additional references accordingly. Specifically, we have included references [14], [16], [30], [31], [36], and [39].